# A Transfer Learning-Based Deep Convolutional Neural Network for Detection of Fusarium Wilt in Banana Crops

Kevin Yan [1,*], Md Kamran Chowdhury Shisher [2] and Yin Sun [2,*,†]

1   Auburn High School, 1701 E Samford Ave, Auburn, AL 36830, USA
2   Department of Electrical and Computer Engineering, Auburn University, Broun Hall, 341 War Eagle Way, Auburn, AL 36849, USA; mzs0153@auburn.edu
*   Correspondence: kyan7472@gmail.com (K.Y.); yzs0078@auburn.edu (Y.S.); Tel.: +1-334-524-1842 (K.Y.)
†   Current address: Department of Electrical and Computer Engineering, Auburn University, 345 W Magnolia Ave, Auburn, AL 36849, USA

**Abstract:** During the 1950s, the Gros Michel species of bananas were nearly wiped out by the incurable Fusarium Wilt, also known as Panama Disease. Originating in Southeast Asia, Fusarium Wilt is a banana pandemic that has been threatening the multi-billion-dollar banana industry worldwide. The disease is caused by a fungus that spreads rapidly throughout the soil and into the roots of banana plants. Currently, the only way to stop the spread of this disease is for farmers to manually inspect and remove infected plants as quickly as possible, which is a time-consuming process. The main purpose of this study is to build a deep Convolutional Neural Network (CNN) using a transfer learning approach to rapidly identify Fusarium wilt infections on banana crop leaves. We chose to use the ResNet50 architecture as the base CNN model for our transfer learning approach owing to its remarkable performance in image classification, which was demonstrated through its victory in the ImageNet competition. After its initial training and fine-tuning on a data set consisting of 600 healthy and diseased images, the CNN model achieved near-perfect accuracy of 0.99 along with a loss of 0.46 and was fine-tuned to adapt the ResNet base model. ResNet50's distinctive residual block structure could be the reason behind these results. To evaluate this CNN model, 500 test images, consisting of 250 diseased and healthy banana leaf images, were classified by the model. The deep CNN model was able to achieve an accuracy of 0.98 and an F-1 score of 0.98 by correctly identifying the class of 492 of the 500 images. These results show that this DCNN model outperforms existing models such as Sangeetha et al., 2023's deep CNN model by at least 0.07 in accuracy and is a viable option for identifying Fusarium Wilt in banana crops.

**Keywords:** convolutional neural network; Fusarium wilt; transfer learning; ResNet-50; banana crop

## 1. Introduction

Banana (*Musa* spp.) is one of the most widely produced cash crops in the tropical regions of the world and the fourth most important crop among developing nations. Over 130 countries export bananas, contributing to a total revenue of 50 billion dollars per year [1]. Fusarium wilt or Panama disease is caused by the *Fusarium oxysporum* f. sp. cubense tropical race 4 (TR4), and is a well-known threat to global banana production [2,3]. TR4 infected hundreds of thousands of hectares of banana plantations throughout countries like China, India, the Philippines, Australia, and Mozambique [4].

Four to five weeks after inoculation with Foc, banana crops begin to exhibit the main symptom of Fusarium wilt, yellowing of their leaves. TR4 can spread through flowing water, farm equipment, infected plant material, and soil contamination. Approximately four to five weeks after being inoculated with TR4, banana crops start displaying the primary symptom of Fusarium wilt: the yellowing of their leaves. These once-vibrant leaves gradually begin to droop and eventually collapse, forming a ring of lifeless foliage encircling the pseudo-stem of the crop [5]. Over time, an increasing number of leaves

experience wilting and collapse, resulting in the entire canopy of the crop being composed solely of withering and deceased leaves.

Currently, the main method used to manage Fusarium wilt is to inspect banana plantations with manual labor in hopes of identifying these yellowing leaves. However, this process requires enormous amounts of time and money to perform [6]. Also, manual inspection quality is affected by the experience and expertise of the farmer performing the inspection, meaning accuracy of diagnosis cannot be ensured [7]. Additionally, there are currently no chemical or physical treatments available that can effectively control Fusarium Wilt. Once the signs of this disease are identified, the only viable treatment option is the rapid removal of the crop in order to prevent a large-scale infection from occurring [8].

Alternatively, Remote sensing has been used in recent studies to detect the presence of Fusarium wilt. For example, a DJI Phantom 4 quadcopter (DJI Innovations, Shenzhen, China) equipped with a MicaSense RedEdge M$^{TM}$ five-band multi-spectral camera to perform remote sensing surveys of banana plantations in China [6]. The multi-spectral images were then used to calculate different vegetation indices, attempting to find the infection status of Fusarium wilt in the plantation. The same approach was used in [9] to detect Fusarium wilt on banana plantations.

The main drawback to these remote sensing detection systems, despite their ability to detect Panama disease, is similar to that of conducting manual inspections: the high cost. The high-quality multispectral and hyperspectral cameras required by these types of detection systems can range in cost from several thousands of dollars to tens of thousands of dollars. Given that Fusarium wilt infections mostly occur in developing countries on the continents of South America, Africa, and South Asia [10], it is clear that farmers in those regions cannot afford high-quality spectral cameras and the computational power used to analyze spectral images.

Presently, convolutional neural networks have been used to detect many different plant diseases through their symptoms within the visible light spectrum. In computer vision, deep convolutional neural networks (CNNs) can achieve excellent performance in image classification tasks [11]. CNNs are a variant of deep neural networks that are designed to mimic the cognitive process of human vision. CNNs receive an input, usually in the form of an image, which is then fed through layers of neurons that perform nonlinear operations, lastly, the output is in the form of a list of scores between 0 and 1, each of which represents the likelihood of the image belonging to an image class. The nonlinear operations at these neurons are optimized through a training procedure [12].

Transfer learning is a technique used in the design of deep learning models to reduce the need for large training datasets and high computational costs for training. Transfer learning essentially works by integrating the knowledge of a previously trained CNN model into a new CNN model designed for a specific task [13]. Transfer learning approaches have been used in plant classification, sentiment classification, software defect prediction and more [14]. There are several pre-trained models that can be selected as the base model for transfer learning, such as ResNet, AlexNet, Inception V-3, VGG16, and ImageNet. In [11], the authors evaluated the aforementioned four pre-trained models in a transfer learning-based plant disease detection task. It was found that ResNet-50 is the best model and achieves an accuracy of 0.9980.

In this paper, we present a transfer learning-based deep CNN model for the detection of Fusarium Wilt in banana crops. The key contributions of this paper are summarized as follows:

- We employed ResNet-50 as the base model, which is accessible in Keras with pre-trained weights on the TensorFlow backend. Originally trained to identify 1000 distinct object classes from ImageNet, the ResNet-50 architecture was adapted for our binary classification task. To do this, we substituted the original final fully connected layer of 1000 neurons with a single neuron fully connected layer to fit our binary output. Then, the ResNet-50 layers were fine-tuned to adapt the knowledge of the base model to the Fusarium wilt detection task. Our CNN model is illustrated in Figure 1.

- The training dataset was obtained from [15], consisting of 72 images of Fusarium wilt-infected banana leaves and 84 images of healthy banana leaves. To augment the dataset size and mitigate overfitting, random flips, rotations, and noise injection, were applied to each image. After training for 100 epochs, the model demonstrated near-perfect performance.
- Furthermore, for testing purposes, we created a new data set from Google searches, comprising 500 distinct images. This testing dataset will provide a robust platform for further research into Fusarium wilt detection.
- Our transfer learning-based CNN model achieved high accuracy and an F-1 score of 0.98, correctly classifying 492 out of 500 images (see Figures 2–5). We also demonstrate the impact of both transfer learning and data augmentation by evaluating our model's performance with and without these techniques. Figure 6 illustrates that our CNN model achieves an accuracy of 0.7685 without transfer learning and data augmentation, while it attains an accuracy of 0.9823 using the transfer learning and data augmentation methods.
- To evaluate the model's discriminative ability on the banana leaf image dataset, we employed a Receiver Operating Characteristic (ROC) curve (refer to Figure 7). The results, depicted by the ROC curve, affirm that our trained model can effectively distinguish between Fusarium wilt-infected and healthy banana plants.
- We conducted a performance comparison between our ResNet-50 model based CNN model and other CNN models: VGG16, AlexNet, and DenseNet121. As illustrated in Figure 8, ResNet-50 exhibited superior performance compared to these models. Therefore, our transfer learning-based ResNet-50 model emerges as a promising candidate for the detection of Fusarium wilt in banana crops.

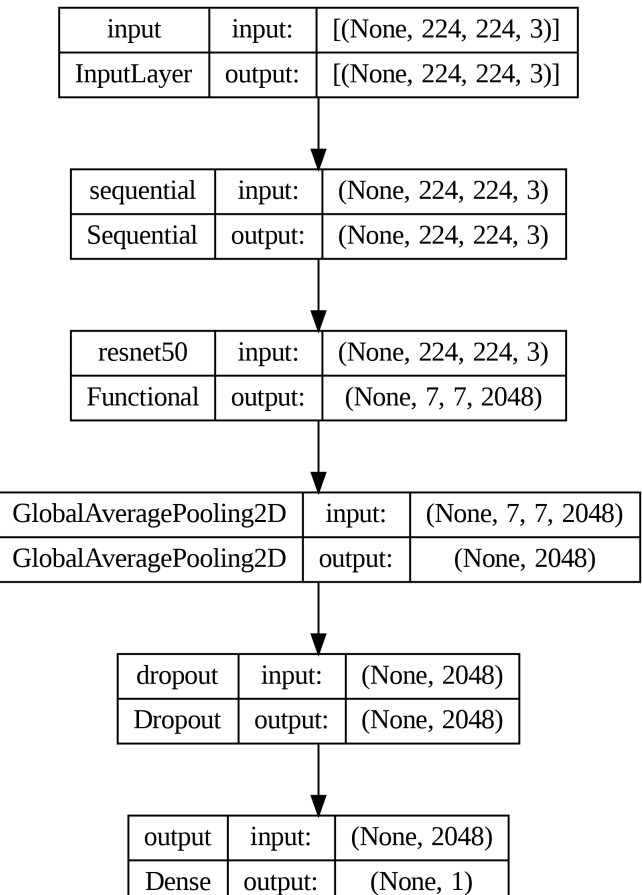

**Figure 1.** The diagram displays the dimensions of the inputs and outputs of each layer of our CNN model along with the names of each layer.

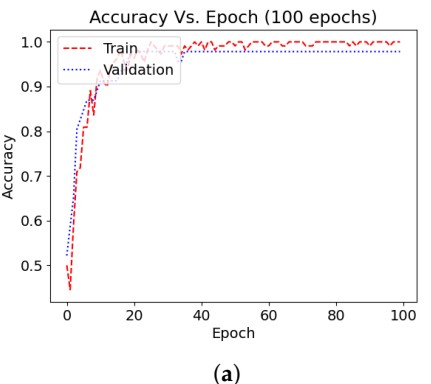
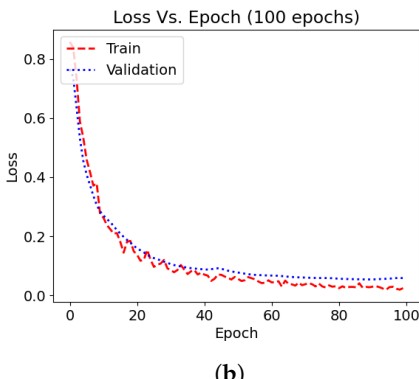

|     (a)     |     (b)     |

**Figure 2.** (**a**) Accuracy (percentage of correctly predicted samples) vs. Epoch and (**b**) Loss (binary cross-entropy loss) vs. Epoch. Visualization of the learning process reveals improvement in both training and validation loss and accuracy for our CNN model with successive epochs.

```
Epoch 1/5
2/2 [==============================] – 55s 49s/step – loss: 0.0050 – binary_accuracy: 1.0000 – val_loss: 0.0232 – val_binary_accuracy: 0.9783
Epoch 2/5
2/2 [==============================] – 1s 285ms/step – loss: 0.0041 – binary_accuracy: 1.0000 – val_loss: 0.0226 – val_binary_accuracy: 0.9783
Epoch 3/5
2/2 [==============================] – 1s 280ms/step – loss: 0.0026 – binary_accuracy: 1.0000 – val_loss: 0.0205 – val_binary_accuracy: 1.0000
Epoch 4/5
2/2 [==============================] – 1s 285ms/step – loss: 0.0024 – binary_accuracy: 1.0000 – val_loss: 0.0182 – val_binary_accuracy: 1.0000
Epoch 5/5
2/2 [==============================] – 1s 298ms/step – loss: 0.0025 – binary_accuracy: 1.0000 – val_loss: 0.0169 – val_binary_accuracy: 1.0000
```

**Figure 3.** These outputs are the values of accuracy and loss of the training and validation sets during fine-tuning.

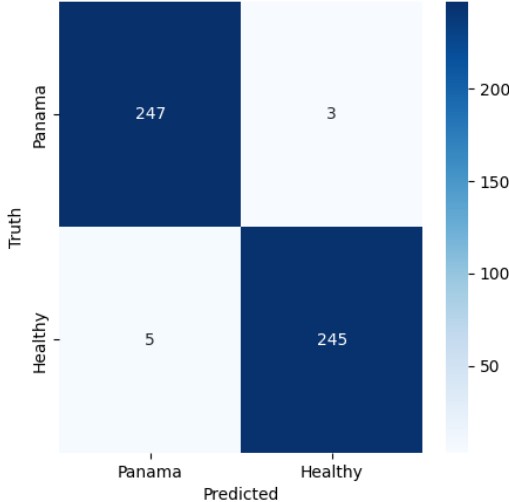

**Figure 4.** The confusion matrix represents the predictions made by the CNN model (500 total predictions).

|              | precision | recall | f1-score | support |
|--------------|-----------|--------|----------|---------|
| Healthy      | 0.99      | 0.98   | 0.98     | 250     |
| Panama       | 0.98      | 0.99   | 0.98     | 250     |
|              |           |        |          |         |
| accuracy     |           |        | 0.98     | 500     |
| macro avg    | 0.98      | 0.98   | 0.98     | 500     |
| weighted avg | 0.98      | 0.98   | 0.98     | 500     |

**Figure 5.** The classification report shown here depicts the precision, recall, F-1 score, accuracy, and the macro and weighted averages of these values.

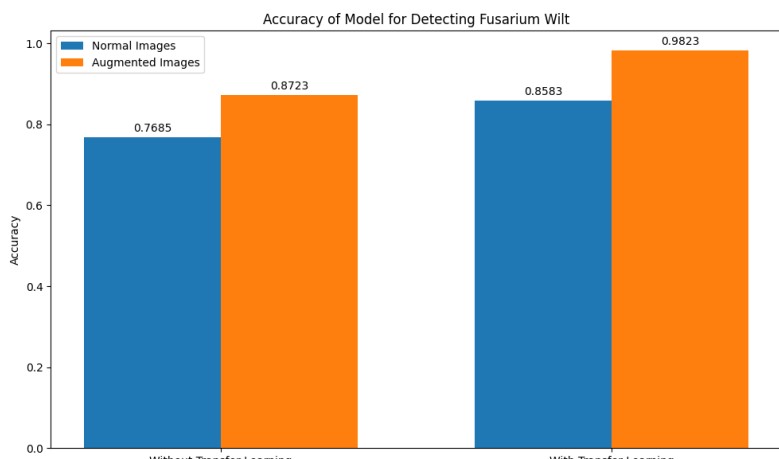

**Figure 6.** The graph shows the different accuracy values as a result of the different evaluation conditions of the model.

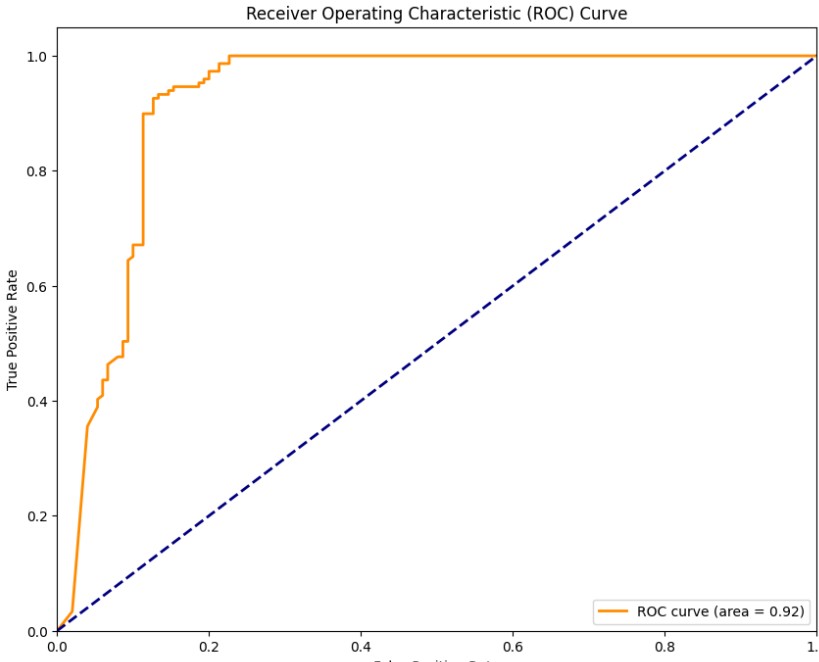

**Figure 7.** ROC curve for the CNN model trained on the Fusarium wilt dataset. The area under the curve (AUC) of 0.92 signifies that the model has good discriminatory power between the positive and negative classes.

### 1.1. Literature Review

CNNs have been employed for the detection of various plant diseases, such as soybean plant diseases [16], apple black rot, grape leaf blight, tomato leaf mold, cherry powdery mildew, potato with early blight, and bacterial spots on a peach [14]. In a study by [17], the authors developed a CNN model capable of detecting 58 classes of plant leaves from diverse crops, including aloe vera, apple, banana, cherry, citrus, corn, coffee, grape, paddy, peach, pepper, strawberry, tea, tomato, and wheat. In another work, Ref. [1] presented a deep learning-based neural network model for identifying Fusarium wilt. Their model achieved an accuracy of 0.9156, evaluated on 700 samples of diseased and healthy banana leaf images. Notably, the accuracy reported in this study surpassed that achieved in [1] by approximately 0.065, reaching 0.98. The study [6] proposed a remote sensing-based detection model for Fusarium wilt on banana plantations in China. Their model utilized changes in vegetation indices caused by Fusarium wilt for disease detection. Meanwhile,

Ref. [18] implemented the Yolo v4 neural network model in a Raspberry Pi for Fusarium wilt detection in banana leaves. The handheld system demonstrated an on-field accuracy of 0.90, offering portability and independence from internet connectivity. The paper [19] introduced a Support Vector Machine (SVM) classification framework for identifying four types of banana diseases in India, including sigatoka, cmv, bacterial wilt, and Fusarium wilt. The SVM classification achieved an average accuracy of approximately 0.85, with accuracy of 0.84, 0.86, 0.85, and 0.85 for the four diseases, respectively.

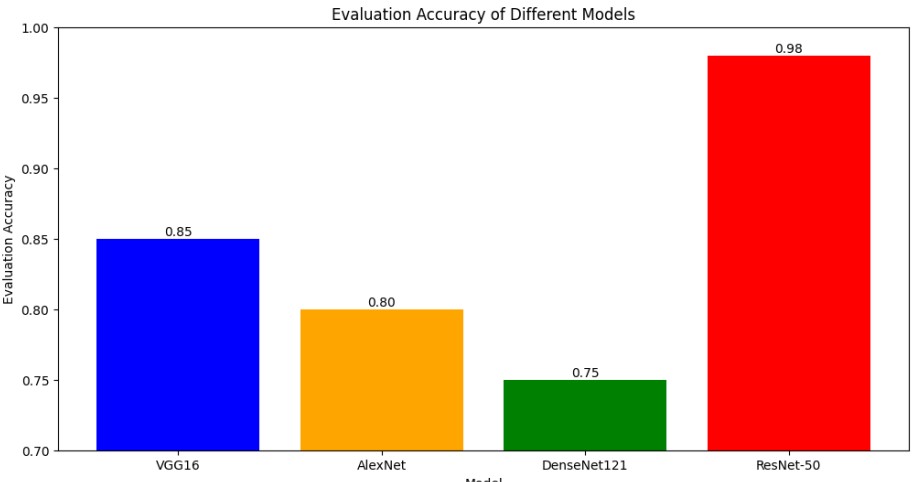

**Figure 8.** Accuracy comparison across different deep learning architectures. ResNet-50 demonstrates superior performance in Fusarium wilt detection.

In a recent development, Ref. [20] conducted an in-depth study leveraging the MobileNetV2 architecture, an efficient CNN model designed for mobile vision applications. In [20], the authors demonstrated the capability of lightweight neural networks when augmented with deep transfer learning techniques to achieve notable accuracy in classifying a diverse array of fruit images, including apples, oranges, and bananas. The practical implications of their study suggest the viability of implementing such models on edge devices within agricultural frameworks. Continuing the exploration of transfer learning potentials, Ref. [21] provided an extensive comparative analysis on the detection of diseases in sunflower crops. The study [21] highlighted the significant performance benefits of applying transfer learning to plant disease detection tasks. By using a CNN pre-trained on extensive datasets and subsequently fine-tuning it for sunflower disease identification, they underlined the robustness of transfer learning in achieving precise disease identification in agricultural applications. In a comprehensive review, Ref. [22] addressed the complications inherent to citrus fruit disease classification. The review [22] critically assessed the methodologies from initial image acquisition to advanced pre-processing and classification via CNN. Similarly, Ref. [23] investigated enhanced methods of data preparation. The study [23] put forth innovative annotation techniques that aid in the efficient training of CNN models for fruit classification tasks, with a specific focus on minimizing manual labeling efforts while enhancing the precision of dataset annotations. In [24], authors presented a novel CNN-based approach for seed classification. In their publication [24], they outlined the development of a CNN model adept at categorizing seeds with substantial accuracy, demonstrating the versatility of CNNs beyond disease identification and into the quality assessment and sorting of agricultural products.

*1.2. Objectives*

The main objectives of this study were to (a) develop and train a transfer learning-based convolutional neural network model for identifying Fusarium wilt in banana crops, (b) utilize a ResNet-50 pre-trained model as the foundational architecture, and (c) evaluate the precision of this model in Fusarium wilt identification.

## 2. Materials and Methods

### 2.1. Dataset

In [15], the authors compiled an extensive dataset featuring diverse images of banana diseases and healthy plant samples. For the purpose of training our CNN model, we utilized a subset of their data that comprised 72 images of Fusarium wilt-infected banana leaves and 84 images featuring healthy banana leaves. The dataset compiled by [15] is accessible at (https://data.mendeley.com/datasets/4wyymrcpyz/1 (accessed on 5 December 2023)) This dataset includes images that allow one to distinguish Fusarium wilt-infected banana leaves from healthy banana leaves. These images were captured at different times of the day and under varying environmental conditions, ensuring a broad range of scenarios for increased model robustness. The original images in the dataset had dimensions of $256 \times 256$ pixels. However, to align with the input layer dimensions of the CNN model, the images underwent resizing to a standardized size of $224 \times 224$ pixels. This resizing process ensured uniform dimensions for all input images, facilitating seamless processing within the CNN model.

### 2.2. Data Pre-Processing and Augmentation

To expand the initial dataset and address the risk of overfitting due to limited data, we employed data augmentation techniques. This involved applying random flipping, rotation, and noise addition to each image in the training dataset. The resulting augmented dataset provided the model with a more diverse and enriched sample for training, enhancing its generalization capabilities beyond the original dataset.

The dataset was divided into two separate subsets: a training dataset and a validation dataset. Specifically, 70% of the dataset was assigned to the training dataset, while the remaining 30% formed the validation dataset. By establishing a clear distinction between the training and validation datasets, the model's performance was effectively assessed and monitored during the training process. The separation facilitated a robust evaluation of the model's capacity to generalize predictions to previously unseen data, ensuring its effectiveness in detecting and classifying Fusarium wilt-infected banana leaves beyond the images it had been trained on.

### 2.3. Transfer Learning

Our study focused on creating a transfer learning-based CNN model specialized in detecting Fusarium wilt in banana crops. We aimed to develop a model that balances computational efficiency with ease of deployment. For this task, we selected Keras [25], a renowned high-level neural network API, in conjunction with TensorFlow [26], a leading machine learning framework. In our search for a suitable pre-trained model, we opted for ResNet-50 [27], a well-established CNN architecture in computer vision. We chose ResNet-50 for its proven effectiveness in image classification tasks and its innovative use of residual blocks to address the vanishing gradient problem in deep networks. The architecture ensures gradient information preservation across layers, enabling optimal weight learning throughout its depth.

We employed a ResNet-50 model pre-trained on the ImageNet dataset with 1000 object categories. Then, we modified its architecture by replacing the original 1000-neuron fully connected layer with a single-neuron layer tailored to our binary classification task.

While newer CNN architectures exist, many of them demand substantial computational resources, potentially hindering deployment in resource-constrained environments. ResNet-50 strikes a balanced trade-off, offering excellent performance without the hefty computational demands of more recent architectures. This aligns well with our goal of creating a model that is powerful and practical for widespread use.

### 2.4. Neural Network Architecture

A CNN model has a sequential structure, stacking each layer on the previous one. This configuration facilitates a systematic flow of information through the network. It also

helps in extracting increasingly abstract features from the input image as it progresses through the layers. Figure 1 illustrates the architecture of our CNN model, depicting the step-by-step transformation of the input image.

### 2.5. Input Layer

Our model initiates with the input layer, which receives an image sized at $224 \times 224$ pixels with three RGB values, forming a 3-dimensional matrix. Each cell in this matrix contains a pixel value representing the intensity of the pixel for grayscale images or, for RGB images, three matrices or channels corresponding to the red, green, and blue components. Subsequently, the input image is fed into the ResNet-50 based layer.

### 2.6. ResNet-50 Layer

ResNet, short for Residual Network, represents a specific type of CNN introduced in [27]. ResNet-50, specifically, is a convolutional neural network with 50 layers (48 convolutional layers, one MaxPool layer, and one average pool layer). Residual neural networks are a subtype of artificial neural networks (ANNs) constructed by stacking residual blocks. The residual block is defined as:

$$y = f(x, W) + x, \tag{1}$$

where $x$, $y$, $W$, and $f(\cdot)$ denote the input, output, weights, and the residual function of the block, respectively. Refer to [27] for detailed information.

In Figure 1, the MaxPool layer and the average pool layer are separated from the ResNet-50 Layer, and details about these two layers are provided below.

### 2.7. Max Pooling Layer

After the 48 convolutional layers of ResNet-50, a max pooling layer is introduced to downsample the feature map. This process, which reduces the spatial dimensions, extracts essential features, enhancing computational efficiency. The downsizing aids in emphasizing the most salient information while discarding irrelevant details.

### 2.8. Global Average Pooling Layer

Once the information passes through the residual blocks and the max pooling layer, a global average pooling layer is applied. This layer reduces the spatial dimensions of the feature map by computing the average value of each feature. By summarizing information across the entire feature map, the global average pooling layer preserves the most relevant features while discarding spatial details. This process focuses on the most descriptive aspects of the image.

### 2.9. Dropout Layer

Following the global average pooling layer, a dropout layer is incorporated into the CNN model. This layer randomly drops out a fraction of the outputs generated by the previous layer during training, serving as a regularization technique. Its purpose is to enhance the model's ability to generalize by reducing the likelihood of overfitting.

### 2.10. Fully Connected Dense Layer

Finally, the CNN model concludes with a fully connected dense layer that conducts linear transformations on the input received from the preceding layers. Because our task (binary classification) is to distinguish between a healthy banana leaf and a Fusarium wilt-infected banana leaf, we include a fully connected output layer with one neuron. This neuron incorporates a sigmoid function, expressed as:

$$\sigma(\mathbf{z}) = \frac{1}{1 + e^{-(\mathbf{W}^T \mathbf{z} + b)}}, \tag{2}$$

where **z** denotes the input vector, the vector **W** contains the weights, and $b$ is the bias.

The sigmoid Function (2) produces a value between 0 and 1. This characteristic allows the model to interpret the output as a probability, indicating the likelihood of a given input belonging to a specific class. For instance, if the output of the last layer exceeds 0.5, we predict the input as a healthy leaf; otherwise, the leaf is Fusarium wilt-infected.

## 3. Results

### 3.1. Training

Prior to training, the ResNet-50 architecture's layers were initialized with weights pre-trained on ImageNet. The weights of the last fully connected layer (i.e., the output layer) were initialized using a uniform random distribution, as detailed in [28]. During training, the pre-trained layers of the ResNet-50 architecture remained frozen, while the output layer was trained using the Adam optimization algorithm with a learning rate of 0.001. The standard binary cross-entropy loss function was employed in training, and two metrics were recorded: *accuracy*, representing the percentage of correctly predicted samples, and *loss*, representing the binary cross-entropy loss.

Let $y_i \in \{0, 1\}$ denote the actual label of the $i$-th input image, where $y_i = 0$ indicates a Fusarium wilt-infected leaf; otherwise, $y_i = 1$ indicates that the leaf is healthy. The output of the neural network is denoted by $p_i$, where $0 \le p_i \le 1$ represents the probability of the $i$-th input being healthy. Let $N$ be the number of images in a dataset. The binary cross-entropy loss (BCE) of the neural network on the given dataset is determined by

$$\text{BCE} = -\frac{1}{N} \sum_{i=1}^{N} y_i \log p_i + (1 - y_i) \log(1 - p_i). \tag{3}$$

The accuracy of the neural network on a given dataset is determined by

$$\text{Accuracy} = \frac{1}{N} \sum_{i=1}^{N} y_i \mathbf{1}(p_i \le 0.5) + (1 - y_i)\mathbf{1}(p_i > 0.5), \tag{4}$$

where $\mathbf{1}(\cdot)$ is an indicator function.

Utilizing our dataset, the CNN model underwent training on the Google Colab platform. The computation was expedited via the cloud-based Graphics Processing Unit (GPU). The model's training consisted of 100 epochs, and the observed metrics exhibited convergence tendencies.

As depicted in Figure 2, post-training, (a) the CNN model registered an accuracy of 100% and a BCE loss of 0.0247 on the training dataset; (b) the CNN model registered an accuracy of 97.83% and a BCE loss of 0.0351 on the validation dataset.

### 3.2. Fine Tuning

After the initial training phase, the last 10 pre-trained layers of the ResNet-50 architecture were unfrozen for model fine-tuning. Then, the weights of all the unfrozen layers of the CNN model were optimized using the Adam optimization algorithm. The learning rate of the model was also adjusted to $10^{-4}$. This is conducted to ensure that the weights of the model are not drastically altered during this process. By fine-tuning, we are essentially adapting the knowledge of the base model to this identification task.

As shown in Figure 3, the outputs of the fine-tuning process show that the model was able to reach a peak accuracy of 100% on both the training dataset and the validation dataset, indicating the pre-trained model's weights having adapted to this classification task successfully.

### 3.3. Model Evaluation

In order to determine the ability of the CNN model to identify Fusarium wilt infections, it was tested on a dataset consisting of 500 images (https://www.dropbox.com/

 (accessed on 5 December 2023)). The images, previously unseen by the model, were taken from a Google search. Using the OpenCV-Python3 library, the data augmentation techniques of rotation, noise addition, and resizing were used to expand the 200 healthy and diseased images to 500 images.

The metrics used to determine the overall performance of the model were: accuracy, precision, recall, and F-1 score. These measures are calculated using Equations (5)–(8), respectively. Accuracy is a straightforward metric that measures the overall correctness of the model's predictions. Precision is a metric that focuses on the positive predictions ($y_i = 1$) made by the model. It is defined as the ratio of true positive predictions (correctly predicted positive samples) to the total number of positive predictions (both true positives and false positives). Recall, also known as sensitivity or true positive rate, measures the ability of the model to correctly identify positive samples. It is defined as the ratio of true positive predictions to the total number of actual positive samples (true positives and false negatives). The F-1 score is the harmonic mean of precision and recall. It provides a balanced measure that considers both precision and recall simultaneously. Additionally, a confusion matrix was visualized through the OpenCV library packages to see the exact number of true and false positive and negative identification classes. It should be noted that the accuracy of random guessing identification for Fusarium wilt will yield an accuracy of 0.50. It should also be noted that there is no method to truly quantify the accuracy of a farmer in identifying Fusarium wilt as this depends on many variables including experience, age, genetics and more.

$$\text{Accuracy} = \frac{\text{TP} + \text{TN}}{\text{TP} + \text{TN} + \text{FP} + \text{FN}}. \tag{5}$$

$$\text{Precision} = \frac{\text{TP}}{\text{TP} + \text{FP}}. \tag{6}$$

$$\text{Recall} = \frac{\text{TP}}{\text{TP} + \text{FN}}. \tag{7}$$

$$\text{F1 Score} = \frac{2 \times \text{Precision} \times \text{Recall}}{\text{Precision} + \text{Recall}}. \tag{8}$$

As depicted in Figure 4, the CNN model with augmented images in the training data and with transfer learning-based design was able to correctly identify 492 out of the 500 total test images to the correct classification class. The eight errors consisted of five false positive and three false negative identifications. The greater number of false positives in this case is considered a more desirable outcome because the farmer can verify that the banana crop is not truly infected before any potential removal of the entire plant. If the number of false negatives was higher, it would pose a larger issue of missing infections, thus, allowing a greater spread of Fusarium wilt.

The classification report shown in Figure 5 depicts the precision, recall, F-1 score, and accuracy of the testing dataset. The CNN model was able to perform with an accuracy of 0.99 on the healthy images and 0.98 on the Fusarium wilt-diseased images. These measures combined with the 0.98 F1 score show the model was able to generalize its prior training to new data. As a comparison for metrics, Ref. [29] trained a deep learning model to identify Fusarium wilt and was only able to achieve 0.9156 accuracy, 0.9161 precision, 0.8856 recall, and 0.8156 score. Their model was also able to substantially outperform existing models with an accuracy nearly 0.20 higher than the second-best model compared in their study. The model in this paper was able to outperform all the metrics of their model; however, this may be due to the original size of the evaluation set being substantially less than the set in their paper. In future real-world testing, this CNN model may yield slightly lower metrics; however, it is still expected that the model will outperform most existing models based on its stellar performance on this dataset.

### 3.4. Comparative Analysis: Impact of Data Augmentation and Transfer Learning

In this section, we analyze the impact of data augmentation and transfer learning on model performance. The evaluation was conducted with four different conditions: (a) model performance without data augmentation and transfer learning, (b) model performance with data augmentation and without transfer learning, (c) model performance without data augmentation and with transfer learning, and (d) model performance with data augmentation and with transfer learning. The results of the evaluations are illustrated in Figure 6.

### 3.5. Receiver Operating Characteristic

A Receiver Operating Characteristic (ROC) curve was also used to evaluate the performance of the model on the same dataset of banana leaf images. The ROC curve is a graphical representation of the true positive rate (TPR) against the false positive rate (FPR) at various threshold settings. It provides an aggregate view of the model's performance across all possible classification thresholds.

$$\text{TPR (Sensitivity)} = \frac{\text{TP}}{\text{TP} + \text{FN}} \tag{9}$$

$$\text{FPR (1 - Specificity)} = \frac{\text{FP}}{\text{FP} + \text{TN}} \tag{10}$$

where:

- TP: True Positives
- FN: False Negatives
- FP: False Positives
- TN: True Negatives

The high AUC value in Figure 7 indicates that the model can distinguish between affected and unaffected plants with high accuracy.

### 3.6. 10-Fold Cross Validation

To enhance the precision and robustness of the Convolutional Neural Network (CNN) model, a 10-fold cross-validation method was implemented. This technique's significance lies in assessing the CNN model's ability to generalize and make accurate predictions on unseen data.

Given a dataset $D$ of size $N$, it is partitioned into 10 mutually exclusive subsets, $D_1, D_2, \ldots, D_{10}$. For each iteration $i$ of the cross-validation: 1. $D_i$ is reserved for validation, and the union of the remaining subsets $D - D_i$ is used for training. 2. The model is trained on $D - D_i$ and validated using $D_i$. 3. Performance metrics, namely training loss, training accuracy, validation loss, and validation accuracy, are computed.

The process is repeated for all subsets, ensuring each data point is used for validation precisely once.

$$\text{Average Validation Accuracy} = \frac{1}{10} \sum_{i=1}^{10} \text{Validation Accuracy}_i \tag{11}$$

$$\text{Average Training Accuracy} = \frac{1}{10} \sum_{i=1}^{10} \text{Training Accuracy}_i \tag{12}$$

ResNet-50's enhanced performance is attributed to its deep layers and the integration of residual blocks. These features help the model recognize complex patterns and avoid issues like vanishing gradients. This evaluation supports the choice of ResNet-50 for tasks requiring high classification accuracy.

An examination of the metrics in Table 1 reveals the model's consistent performance across distinct subsets. The gradual rise in training accuracy denotes the CNN's adaptative learning capability. In parallel, the upsurge in validation accuracy indicates proficient

generalization on novel data vectors. This systematic 10-fold cross-validation solidifies the CNN model's efficacy in detecting Fusarium wilt, positioning it as a pivotal tool for disease identification and mitigation in agriculture.

**Table 1.** Metrics for the 10-fold cross-validation of the CNN model, segmented by subset.

| Subset | Training Loss | Training Accuracy | Validation Loss | Validation Accuracy |
|--------|--------------|------------------|-----------------|--------------------|
| 1 | 0.7750 | 0.5188 | 0.9880 | 0.3125 |
| 2 | 0.6890 | 0.6250 | 0.8267 | 0.3750 |
| 3 | 0.7107 | 0.5813 | 0.6750 | 0.5312 |
| 4 | 0.5410 | 0.7375 | 0.5630 | 0.7188 |
| 5 | 0.3574 | 0.8627 | 0.4693 | 0.7812 |
| 6 | 0.2753 | 0.9500 | 0.4069 | 0.8438 |
| 7 | 0.2729 | 0.9375 | 0.3536 | 0.8438 |
| 8 | 0.2361 | 0.9688 | 0.3086 | 0.8750 |
| 9 | 0.2406 | 0.9125 | 0.2694 | 0.9688 |
| 10 | 0.1607 | 0.9804 | 0.2367 | 0.9688 |

*3.7. Comparative Evaluation between Architectures*

We assessed the performance of the ResNet-50 model in detecting Fusarium wilt and compared it with other CNN architectures: VGG16, AlexNet, and DenseNet121. All models were trained using the same dataset to ensure consistent evaluation conditions.

As presented in Figure 8, ResNet-50 achieved an accuracy of 0.98 on the testing dataset, outperforming the other architectures. Each model was set up using standard configurations to maintain fairness in the comparison.

## 4. Discussion

The accurate detection of Fusarium wilt in banana leaves using computer vision is a central challenge in agricultural tech research. Traditional methods often encounter difficulties in consistently differentiating between healthy and diseased banana leaves, particularly when symptoms are subtle. This study explored the application of transfer learning and deep convolutional neural networks to address this challenge.

The model introduced in this research, based on the CNN architecture, demonstrated commendable efficacy, surpassing several contemporary methods. In performance metrics, this model achieved an ROC AUC of 0.92, positioning it favorably against numerous existing algorithms. Such accuracy was achieved within a brief training duration. Training encompassed a mere 100 epochs and concluded within an hour, highlighting the efficient computational capabilities of the Google Colab GPU. An initial dataset of 156 images served as the starting point. Through data augmentation techniques, this dataset was expanded. The augmented dataset enabled the model to effectively differentiate between healthy and wilt-afflicted banana leaves.

When benchmarked against other solutions available, the model shows significant promise for integration into platforms like Unmanned Aerial Vehicles (UAVs) or mobile applications. Such integrations can significantly enhance agricultural practices by providing advanced tools for early Fusarium wilt detection and consequently reducing crop loss.

Future perspectives in computational research suggest the emergence of even more streamlined models tailored for real-world applications. For optimal performance, data collection from specific plantations remains a crucial aspect. Such targeted datasets ensure that models can cater to diverse environmental conditions specific to different agricultural regions. Techniques like federated learning can further optimize these models, allowing for on-field adjustments in alignment with unique crop conditions.

The potential of Generative Adversarial Networks (GANs) also warrants attention. GANs can aid in the generation of synthetic training datasets that mirror real-world conditions, thus enhancing the training process.

In summary, this research highlights the capabilities of CNNs and transfer learning in the realm of agricultural technology, offering a foundation for further advancements in the detection and management of crop diseases.

**Author Contributions:** Conceptualization, K.Y.; methodology, K.Y.; software, K.Y. and M.K.C.S.; validation, K.Y., M.K.C.S. and Y.S.; formal analysis, K.Y., M.K.C.S. and Y.S.; investigation, K.Y.; resources, K.Y., M.K.C.S. and Y.S.; data curation, K.Y.; writing—original draft preparation, K.Y.; writing—review and editing, K.Y., M.K.C.S. and Y.S.; visualization, K.Y. and M.K.C.S.; supervision, K.Y., M.K.C.S. and Y.S.; project administration, Y.S.; funding acquisition, Y.S. All authors have read and agreed to the published version of the manuscript.

**Funding:** This research was supported in part by the NSF under grant no. CNS-2239677 and by the USDA-NIFA, AFRI Competitive Program, Agriculture Economics and Rural Communities, under grant no. 2023-69006-40213.

**Data Availability Statement:** This study utilizes a modified data set from previous work (Medhi and Deb, 2022) for training neural networks. The data set used in evaluating the trained neural network is accessible via the following link: https://www.dropbox.com/scl/fo/uvcgie7p8izwwqf3norp7/h?rlkey=7066yyc9w5ezcju4m3gu594uc&dl=0 (accessed on 5 December 2023). We used the dataset compiled by [15] to train the neural network model. Any additional data not included in the main manuscript can be obtained from the corresponding author upon reasonable request.

**Conflicts of Interest:** The authors have no known competing financial or non-financial interests that are directly or indirectly related to the work submitted for publication.

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
