# Peer review of "A Transfer Learning-Based Deep Convolutional Neural Network for Detection of Fusarium Wilt in Banana Crops"

_agriengineering, doi:10.3390/agriengineering5040146_

Round 1

Reviewer 1 Report

Comments and Suggestions for Authors

This paper detects the Panama disease in Banana trees. It is a well-written and detailed paper. However, the below comments will strengthen the paper:

1. Add ROC Plots.

2. Also, discuss the reason for selecting ResNet-50. There are a lot of newer CNN architectures. Why did you select an old architecture?

3. As the work is very straightforward and simple, add the 10-fold cross-validation to show the robustness of the model. 

4. Basic punctuation and grammar checks are also needed.

5. Check all the references for format as several of them have missing information e.g. reference no. 4. Accessed data is shown in the template, not the date accessed. 

Comments on the Quality of English Language

Writing is compact and easy to read.

Reviewer 2 Report

Comments and Suggestions for Authors

see the attachment 

Comments on the Quality of English Language

Moderate editing of English language required

Round 2

Reviewer 1 Report

Comments and Suggestions for Authors

All the comments were addressed. I recommend accepting the paper.

Author Response

Thank you for your support of our paper. 

Reviewer 2 Report

Comments and Suggestions for Authors

author has incorporated some of the comments but there are some comments which have not been answered which as as follows:

- the reference style is mentioned in APA in the text whereas in the reference list the references are given in numbering. Author must use numbering style as well in the text. 

The results and contributions presented in this article raise substantial concerns. Upon careful examination, it appears that the authors' input into this research is limited. They have employed a publicly available dataset, utilized a pre-trained ResNet model without significant modification, and trained it on a comparatively small dataset. The primary contribution highlighted in this paper appears to be the generation of 400 augmented available tools and techniques, potentially diminishing its significance as a unique contribution. As a result, it is imperative for the authors to clarify and emphasize the original aspects or novel methodologies employed in their work, as the current state of the research may not sufficiently demonstrate a substantive contribution to the field of study. 

--       When it comes to related work, it contains limited information. Author must add more recent works about deep learning in agriculture. Some of the studies are as follows, author should include these in literature and highlight their contribution in the literature. Fruit Image Classification Model Based on MobileNetV2 with Deep Transfer Learning Technique; Harnessing the Power of Transfer Learning in Sunflower Disease Detection: A Comparative Study; Image Acquisition, Preprocessing and Classification of Citrus Fruit Diseases: A Systematic Literature Review; Enhancing image annotation technique of fruit classification using a deep learning approach; A convolution neural network-based seed classification system.

To demonstrate a meaningful contribution, the author should consider the following suggestions:

-       Model Modification: Rather than solely changing the last layer of the ResNet model from 1000 to 2 nodes, the author should consider more substantial modifications or adaptations to the architecture that are specific to the problem at hand. Explain how these modifications enhance the model's suitability for the task of detecting Fusarium Wilt.

-       Performance Comparison: To showcase the effectiveness of transfer learning, provide a comparison of the model's performance before and after transfer learning. This will help highlight the improvements achieved through the transfer learning process.

-       Data Augmentation Evaluation: Present a performance comparison between the model trained without data augmentation and the model trained with data augmentation. Demonstrating the impact of augmentation on model performance is essential to justify its use.

-       Epochs and Training Duration: Extend the training duration beyond 15 epochs to demonstrate the model's convergence and performance stability. Training for at least 100 epochs, as suggested, would be more representative of the model's capabilities.

-       Publication Readiness: Without implementing these modifications and providing comprehensive performance evaluations, the study may not be suitable for publication in its current form. Addressing these issues will enhance the research's credibility and significance.

Comments on the Quality of English Language

Minor editing of English language required

Author Response

Our responses to the points are in the document attached.
